# Historical Perspective: Models of Parkinson’s Disease

**DOI:** 10.3390/ijms21072464

**Published:** 2020-04-02

**Authors:** Shyh Jenn Chia, Eng-King Tan, Yin-Xia Chao

**Affiliations:** 1National Neuroscience Institute, Singapore 308433, Singapore; chia.shyh.jenn@sgh.com.sg; 2Department of Neurology, Singapore General Hospital, Singapore 169856, Singapore; 3Duke-NUS Medical School, Singapore 169857, Singapore

**Keywords:** Parkinson disease models, neurotoxic models, genetic models, advantages, limitations

## Abstract

Parkinson’s disease (PD) is the most common movement disorder with motor and nonmotor signs. The current therapeutic regimen for PD is mainly symptomatic as the etio-pathophysiology has not been fully elucidated. A variety of animal models has been generated to study different aspects of the disease for understanding the pathogenesis and therapeutic development. The disease model can be generated through neurotoxin-based or genetic-based approaches in a wide range of animals such as non-human primates (NHP), rodents, zebrafish, *Caenorhabditis* (*C.*) *elegans*, and drosophila. Cellular-based disease model is frequently used because of the ease of manipulation and suitability for large-screen assays. In neurotoxin-induced models, chemicals such as 6-hydroxydopamine (6-OHDA), 1-methyl-4-phenyl-1,2,3,6-tetrahydropyridine (MPTP), rotenone, and paraquat are used to recapitulate the disease. Genetic manipulation of PD-related genes, such as α-Synuclein(SNCA), Leucine-rich repeat kinase 2 (LRRK2), Pten-Induced Kinase 1 (PINK1), Parkin(PRKN), and Protein deglycase (DJ-1) Are used in the transgenic models. An emerging model that combines both genetic- and neurotoxin-based methods has been generated to study the role of the immune system in the pathogenesis of PD. Here, we discuss the advantages and limitations of the different PD models and their utility for different research purposes.

## 1. Introduction

Parkinson’s disease (PD) is a chronic neurodegenerative disease named after James Parkinson who reported the clinical syndrome more than two centuries ago. Pathologically, PD is the result of selective degeneration of dopaminergic neurons in the Substantia Nigra, which causes decreased level of dopamine in the striatum and lead to abnormal motor control [1]. The motor symptoms include bradykinesia, muscle tone rigidity, resting tremor, and postural instability. In addition, PD patients also display several non-motor symptoms, such as sleep disorders, dementia, sensory abnormalities, and autonomic dysfunctions [2]. The etiology of PD is still largely unknown, although research suggests that the cause involves multifactorial factors such as genetics, environmental agents, and aging [3].

The current treatment regime uses levodopa or dopaminergic agonists to target the primary pathophysiology of PD, which only alleviates the motor symptoms by restoring neurotransmission [4]. However, long term use of these drugs can lead to severe side effects such as wearing-off symptoms and other motor complications [5,6]. When the fluctuation begins, other dopaminergic medications such as monoamine oxidase type B inhibitors, dopamine receptor agonists and N-methyl-d-aspartate (NMDA) receptor antagonist amantadine are used in conjunction to manage the levodopa induced complications [7]. Post-mortem neuropathological confirmation is the gold standard for confirming the diagnosis. The lack of access to human brains have led scientists to develop diverse range of experimental models using animals and in vitro cultured cells that could mimic different aspects of PD.

## 2. Parkinson’s Disease (PD) Models

Three major animal groups have been extensively used for modeling PD: rodents, non-human primates (NHP), and non-mammalian species. Moreover, cell-based model has also emerged as a feasible disease model for PD. There are distinct advantages and limitations of each group which determine the suitability of the model for a specific experiment.

## 3. Rodents

Rodents are commonly used as the animal model because of the ease of care in laboratory environment and availability of transgenic mouse strain as well as established protocols. Specifically, rodents are used to model PD because the nigrostriatal dopaminergic degeneration directly correlates with motor deficits observed in these animals. The behavioral phenotype of rodent unilateral lesions can be examined through a series of tests including measuring movement, grip, or strength of the front paws. The most common behavioral test for PD includes the open field test for locomotor activity, pole test to measure bradykinesia, and stepping test to measure akinesia. Other motor tests including grip strength test, grip coordination test, and rotarod test measure the strength, coordination, and balance. For dyskinesia, it can be measured through tremor monitor and posture assessments [8,9].

The non-motor symptoms are often described in rodents with partial nigrostriatal lesions. The non-motor assessments include activity (sleeping, drinking, and eating) monitoring, tail suspension test, and forced swim test to assess weight loss, depression, and behavioral despair, respectively [10,11]. Other aspects of non-motor symptoms can also be examined through monitoring, for instance presence of excessive grooming due to compulsive behavior and reduction in nest building behavior due to lacking motivation in goal oriented task [12].

## 4. Non-Human Primates (NHP)

In medical research, NHP has played a critical role in providing significant insights into the mechanism of disease because NHP is closely related to humans genetically and physiologically [13]. However, studies of NHP require extensive labor and expenses as well as ethical considerations. To date, only an estimated 10% of PD research are carried out in NHP. Due to the high cost and ethical issues, NHP studies are often performed for preclinical evaluation of therapies [14]. The parkinsonian symptoms in NHP are generated through administration of neurotoxin or viral vector carrying abnormal genes. The severity of the phenotype can be measured through a Unified Parkinson’s Disease Rating Scale (UPDRS)-like measure, however this assessment on NHP has not been standardized [15].

NHP models exhibit symptoms like humans, for instance macaques show Levodopa-induced dyskinesia that resemble human chorea and dystonia. Akinesia and axial rigidity could be assessed in NHP through jumping in the tower test and hourglass test, respectively [16]. In addition, it has been suggested that macaques better replicate human sleeping pattern as compared to rodents, which makes it a superior model to study sleep or social behavior related symptoms [17]. Moreover, neuroimaging studies have been done in NHP and that has provided valuable information when compared to patients in clinical studies [18]. Hence, NHP is a valuable animal model for PD but the study requires high resource support, expertise, and is time consuming.

## 5. Non-Mammalian Species

Small organisms such as *Caenorhabditis* (*C.*) *elegans*, drosophila, and zebrafish have been used because of the ease of genetic manipulations, short life cycle, low costs of maintenance, and most importantly well-defined neuropathology and behavior [19]. Hence, it is particularly useful for large scale drug screening. *C. elegans* is a 959-cell nematomorph that has a simple nervous network comprising 302 neurons and a dopaminergic network with exactly 8 neurons [20]. *C. elegans* expresses various human gene homologues of LRRK2 (*lrk-1*), PINK1 (*pink-1*), Parkin (*pdr-1*), and DJ-1 (*dnaj-1.1* and *dnaj-1.2*) that are directly implicated in familial forms of human PD but do not express α-Synuclein [21].

In Drosophila, the pathway of dopamine synthesis is similar to humans. Thus, Drosophila has emerged as a feasible model for studying neurodegeneration in PD. Neurotoxin-treated drosophila exhibits PD-like symptoms including degeneration of dopaminergic neurons, locomotor defects and oxidative stress [22]. Zebrafish has been widely used in the study of PD pathogenesis because of rapid life cycles and close genetic similarity to human. Additionally, the dopaminergic neurons in zebrafish’s posterior tuberculum (homologous to SN in human) are well characterized. When exposed to neurotoxin, zebrafish displayed alteration in locomotor activity [23]. These findings suggest that the non-mammalian species are feasible model systems to study the molecular mechanism of the disease.

## 6. Induced Pluripotent Stem Cell (iPSC)-Derived PD Model

Recently, Yamanaka et al. made a significant breakthrough by inducing pluripotent stem cells. They generated induced pluripotent stem cell (iPSC) by overexpressing four major transcription factors, Oct3/4, c-Myc, Sox2, and Klf4 [24]. iPSC-derived PD models offer a unique advantage as experiments could be performed directly on the cells isolated from patients. Studies using patient derived iPSCs with certain mutations have certainly provided significant insights on the molecular pathology of PD [25,26]. However, current neuronal cultures lack the complete physiological network of connections that mimic the brain physiology. PD cellular models are useful for researchers to focus on one aspect of the disease and it has been known that the manipulation of cellular models is relatively quick, and more cost effective compared to animal models. Cellular models are ideal for large scale drug screening that could help narrow down potential drug targets for further validation in animal models [27].

## 7. Mechanisms of PD Models

In general, current PD experimental models are categorized into two main groups: neurotoxins and genetics. Neurotoxin-based models induce the rapid degeneration of nigrostriatal dopaminergic neurons which mimic the sporadic PD. The neurotoxin-based models could be developed by introducing neurotoxins such as 6-hydroxydopamine (6-OHDA), 1-methyl-4-phenyl-1,2,3,6-tetrahydropyridine (MPTP), paraquat, and rotenone. With the addition of neurotoxins, oxidative stress is generated and this can lead to cell death in dopaminergic (DA) neuronal population. However, the disadvantage is lack the formation of Lewy bodies the main pathology hallmark of PD. Despite the limitations, these animal models have contributed significantly to discovering the disease processes and potential therapeutic targets in PD [28].

In recent studies, rare forms of PD have been found to be associated with genetic mutations in α-Synuclein, parkin, LRRK2, PINK1, or DJ-1, which could potentially be the therapeutic targets. Genetic models of PD are created through transgenic overexpression of α-Synuclein and LRKK2 or knockout/knockdown for genes such as Parkin, DJ-1, and PINK1, to study the molecular mechanism of these genes in PD pathology [29]. Genetic model is a compelling approach to examine the association of specific mutations in the familial forms of PD. However, few of these models reproduce the complete features of the disease and often quite different from the human conditions. For instance, most of the genetic models failed to induce significant loss of dopaminergic neuron, the main pathological hallmark of PD [30].

### 7.1. Neurotoxin-Induced Animal Models

It has been known that dopaminergic neurons lesion could be induced through administration of the structural analogs of dopamine, such as 6-OHDA and MPTP. In addition, chronic exposure to agricultural chemicals has been reported to have harmful effect to the neurons and increase the risk of PD. These findings allow research groups to generate different PD models using neurotoxins.

#### 7.1.1. 6-Hydroxydopamine (6-OHDA)

The first animal model of PD was generated based on the administration of 6-OHDA. It has been widely used in PD research because of its consistent behavioral phenotype in these models and predictable degeneration in dopaminergic neurons. Intracerebral injection of 6-OHDA is required because it does not penetrate the blood brain barrier [31,32]. Once inside the cell, 6-OHDA rapidly oxidizes and produces reactive oxygen species such as hydrogen peroxide, superoxide radicals, and hydroxyl radicals that lead to mitochondrial dysfunction. Administration of 6-OHDA in different brain regions will cause distinctive pattern of neurons degeneration. For instance, 6-OHDA injection directed to striatum will damage the axon terminals first in striatum followed by dopaminergic neuron degeneration in the substantia nigra (SN) [33]. In contrast, injection of 6-OHDA into SN leads to massive destruction of dopaminergic neuron. Hence, the latter approach shows relatively severe symptoms.

Numerous studies have been conducted on 6-OHDA models to examine the neuroprotective function of some compounds. For instance, combined treatment of antioxidants and iron chelators have shown positive effects in neutralizing 6-OHDA neurotoxicity [34]. In addition, therapeutic effect of Tanshinone I have also been examined using the 6-OHDA model [35]. It is known that 6-OHDA model does not mimic the pathology of PD perfectly which lacks the Lewy Bodies (LB) formation [36].

#### 7.1.2. 1-Methyl-4-phenyl-1,2,3,6-tetrahydropyridine (MPTP)

MPTP is one of the most used neurotoxins in PD animal models. MPTP is a lipophilic molecule which makes it able to cross the blood brain barrier easily. After systemic administration, MPTP can be oxidized into potent dopaminergic neurotoxin 1-methyl-4-phenylpyridinium ion (MPP+) by the monoamine oxidase B in astrocytes. MPP+ is a toxic metabolite that is readily absorbable by dopaminergic neuron through dopamine transporter (DAT) because of its structural similarity to dopamine [37]. Subsequently, MPP+ induces progressive loss of DA neurons in the SN together with decrease striatal dopamine levels [38]. The mechanism of cell death is induced by MPP+ inhibition of complex I in mitochondrial respiration. This results in rapid decrease in adenosine triphosphate (ATP) concentration in striatum and SN followed by apoptosis and necrosis of DA neurons [39].

MPTP animal model is recommended for studying mitochondrial dysfunction in PD because of its unique antagonist activity in mitochondria. To closely reproduce PD in animal models, chronic administration of the toxin needs to be carried out over weeks which will result in continuous cell death after completion of toxin administration. The topographic pattern of dopaminergic cell loss in the striatum does replicate that of PD [40]. However, MPTP model lacks the most important neuropathological feature of PD which is the formation of Lewy Bodies (LB) [36].

### 7.2. Paraquat

Paraquat is a widely used herbicide that was identified as a neurotoxicant based on its structural similarity to MPP+. The use of paraquat as common pesticide has raised public concern because it could be an environmental contributor to the etiology of PD [41]. The toxicity of paraquat to neural system was tested in frog few decades ago. The result showed that cumulative paraquat dosing has induced several behavioral features as observed in PD [42]. Additionally, systemic injection of paraquat in mice has produced loss of dopaminergic neurons. Paraquat reportedly cross the blood brain barrier through a neutral amino acid carrier. Though Paraquat has structural similarity to MPP+, it does not inhibit mitochondrial complex I like MPP+. Instead, Paraquat impairs the redox cycling of glutathione and thioredoxin. This will affect the function to protect against oxidative stress in cells [43].

In the characterization of the paraquat model of PD, nigral dopaminergic neuron loss was observed without striatal dopamine depletion. This indicates that the model represents certain pathological features of clinical PD but may have different neurochemical effects. However, a study has shown that chronic administration of paraquat resulted in chronic neurodegeneration and dopamine depletion that could be used to study preclinical stage of PD [44]. Additionally, synergistic combinations of paraquat with other compounds have been adopted to produce PD phenotype in animal. For instance, experiments combining neonatal iron exposure with adult paraquat exposure have been conducted and demonstrated age-dependent nigral dopamine cell loss [45]. Based on the data, paraquat PD model could be useful to study the earlier stages of the disease compared to other models because the PD phenotype develops chronically.

### 7.3. Rotenone

Rotenone is a natural compound found in plants that have been used as insecticides. It is lipophilic and can penetrate the blood brain barrier to inhibit mitochondrial complex I like MPTP. However, Rotenone produces systemic inhibition that is different from MPTP which targets on the catecholaminergic neurons [46]. The first Rotenone model was created through stereotaxic injection into the parenchyma at extremely high concentration, which resulted in significant decreases in striatal dopamine and serotonin [47]. However, it is believed that the results produced by such high doses are not specific for dopaminergic neurons. Indeed, high concentration of rotenone has induced liquefactive necrosis in striatal region. In contrast, when Rotenone is administrated chronically at a lower dose, it has induced selective cell degeneration at nigrostriatal region [48].

In rotenone animal model, α-Synuclein inclusions like Lewy body were observed in the surviving dopaminergic neuron [48]. Additionally, other PD related features were observed in rotenone animal model such as motor deficits, catecholamine depletion, and nigral dopamine cell loss. With its ability to reproduce key pathological features of human PD, it can be a valuable tool to conduct studies on neuroprotection. This model has been used in a research studies that examined the efficacy of melatonin in neuroprotection [49].

## 8. Genetic Models of PD

Genetics plays a significant role in PD pathogenesis. Research studies have identified the disease-causing gene using linkage analyses and association analyses for familial PD and idiopathic PD, respectively [50]. *SNCA* (α-Synuclein) is the first gene identified in familial PD [51]. Following the breakthrough, several other familial PD-linked genes were discovered such as parkin, DJ-1, PINK1, and LRRK2 [52,53,54,55].

### 8.1. α-Synuclein

α-Synuclein is a small protein (at size of 14 kDa) that is present abundantly in the pre-synaptic terminals. The specific physiological function of α-Synuclein remains uncharacterized. However, it has been known to be associated with the regulatory activity in membrane and vesicular dynamics [56,57]. Mutations of α-Synuclein have been characterized in the inherited form of PD, for instance substitutions (A53T, A30P, and E46K), duplication, or triplication [58]. Importantly, α-Synuclein is identified as the major building block of Lewy Bodies (LBs). Based on the significant findings, scientists have started to model PD via the over expression of wild type or mutant forms of α-Synuclein in animals. The first ever transgenic mouse model was created by Masliah et al. [59]. They have observed progressive formation of neuronal inclusions in the hippocampus, SN, and neocortex that antibodies stained positively to α-Synuclein. However, the inclusions are not of fibrillar structure that resembles LBs. In addition, there was no significant dopaminergic neuron degeneration observed in the mice [59]. The first transgenic mouse model generated by the group did not fully replicate human pathology.

Following that, more α-Synuclein models were generated through expression via the tyrosine hydroxylase (TH) promoter. This is to achieve localized effect of α-Synuclein which expresses specifically in catecholaminergic neurons [60]. Similarly, this approach has failed to induce α-Synuclein inclusions and dopamine cell loss. However, in a model with double mutant (A30P/A53T), apparent neurites dystrophy was reported. This phenotype was accompanied by deterioration in motor activity and neuronal aggregates [61]. These models are useful to elucidate the function of α-Synuclein related neurodegeneration, though the clinical relevance of this model for PD is questionable.

### 8.2. Leucine-Rich Repeat Kinase 2 (LRRK2)

LRRK2 mutations are associated with autosomal dominant pattern of inheritance in familial PD, with varying penetrance in different population. G2019S and R1441C/G are the two most common LRRK2 mutations [62]. Similarly, most LRRK2 transgenic animal models failed to recapitulate the important PD hallmarks. Interestingly, BAC-LRRK2-R1441G transgenic mice show motor deficits and axonal pathology in the striatum, however in the absence of clear dopamine cell loss and α-Synuclein aggregations [63].

Other LRRK2 models have been created using viral vectors such as herpes simplex virus (HSV) and adenoviral vectors. Transfection of LRRK2-G2019S is known to be more effective than WT LRRK2 in stimulating neurodegeneration and formation of inclusions. Infection of HSV-LRRK2-G2019s in mouse striatum has achieved approximately 50% degeneration of the dopaminergic neuron in the SN [64]. LRRK2 models could provide valuable information on the association of genetic mutations and environmental factors as well as to unravel the complicated mechanisms behind their functions in PD. The effect of LRRK2 kinase inhibition has been tested and recommended as the potential therapeutic option in PD [65].

### 8.3. Parkin

Parkin is the most common autosomal recessive mutations found in the early onset of PD, which is linked to approximately 50% and 20% of familial and idiopathic cases respectively. Parkin is a ubiquitin ligase that plays an important role in proteasomal degradation and the disease-causing mutations of parkin has resulted in loss of function [66]. The Parkin-linked parkinsonism could lead to the accumulation of neurotoxic substrate. Numerous knockout Parkin models have been generated but with no success. None of the knockout models recapitulate typical phenotype of PD [67].

Nonetheless, pathophysiologic studies of Parkin and its role in neuroprotection have been widely reported. Primary midbrain culture prepared from the Parkin knockout lines exhibit increased susceptibility to the insult by rotenone. Further supporting this, overexpression of parkin has prevented dopaminergic neurodegeneration in rats treated with 6-OHDA as well as mice treated with MPTP [68]. In summary, Parkin model is still far from being ideal animal model for PD, however Parkin itsef could be a potential therapeutic target.

### 8.4. Protein Deglycase (DJ-1)

DJ-1 mutations have been known to be associated with recessive forms of familial parkinsonism. Several studies suggest that DJ-1 is an antioxidant protein, which is useful in counteracting the oxidative environment of DA neurons [69]. Unfortunately, elimination of DJ-1 in mice did not induce dopamine cell loss in the SN even at the old age. Moreover, no accumulation of inclusion bodies detected in these mice. DJ-1 knockout lines were tested with reduced DA release in the striatum as a result of DA reuptake by the DA transporter (DAT), however it is not observed in all knockout line. DJ-1 null mice display hypokinesia in the open field tests [70]. The observations suggest that DJ-1 deficiency could trigger motor dysfunction even in the absence of nigral neurodegeneration [71,72]. However, due to limited pathology data from DJ-1 data, it is difficult to conclude whether DJ-1 null model mirrors the human condition.

It is important to note that DJ-1 knockout mice model is hypersensitive towards the insults of MPTP and rotenone. Locomotor of mouse model remains unaffected in the absence of DJ-1. However, apparent locomotor deficit was observed in DJ-1 knockout mice upon neurotoxin treatments as compared to WT mouse model. This indicates that DJ-1 plays a neuroprotective role in our neural system [73]. DJ-1 knockout mice could be a valuable tool to study the molecular mechanism of PD.

### 8.5. Pten-Induced Kinase 1 (PINK1)

Mutations in Pten-induced kinase 1 (PINK1) are associated with recessive parkinsonism. PINK1 is a neuroprotective kinase found primarily in the mitochondria and cytosolic compartments and plays a role in neuronal differentiation [74]. Increased expression of PINK1 induced neurite outgrowth in SH-Sy5y cells and the length of dendrites in dopaminergic neurons [75]. In addition, studies have shown that PINK1 mutants display dopaminergic neuronal degeneration together with locomotive defects. Phenotype of PINK1 mutants has been characterized using transmission electron microscopy analysis and a rescue experiment which showed that mitochondrial dysfunction was responsible for all phenotypes observed in PINK1 null drosophila [76].

Both PINK1 and parkin mutants have been found to have similar phenotypes including abnormally positioned wings, disorganized muscle fiber with enlarged mitochondria, impaired flight ability, and reduced climbing rate [77]. Thus, PINK1 and parkin possibly play important roles in the same pathway. Rescue experiments were conducted in PINK1 null flies and showed that Parkin work downstream of PINK1 in the same pathway [76].

PINK1 knockout mouse models have also been generated, however there is no obvious PD pathology observed in the brain [72,78]. Studies have shown that PINK1 knockout mice were susceptible to oxidative stress and ROS production though without dopaminergic neuronal degeneration or reduced striatal dopamine levels. Complete suppression of PINK1 function in G309D-PINK1 transgenic mice can induce an age-dependent dopamine reduction with decreased locomotor activity [78].

## 9. Emerging Models

In vitro midbrain organoids have been generated from iPSC with different protocols [79,80,81]. The midbrain organoids generated from the PD patients iPSCs or genetically modified healthy controls iPSC may reflect the disease progress in human patients [82]. Mechanisms underlying dopaminergic neuron demise and potential drug treatments can be tested in these models.

Recent findings of clinical association of HLA with PD and potential involvement of immune system in the pathogenesis have shed light on a new avenue of the disease [83,84]. Protocols inducing midbrain inflammation or autoimmunity which may lead to dopaminergic neuron degeneration will provide novel animal models to understand the role of immune system in PD pathogenesis and treatment [85].

## 10. Conclusions

Various neurotoxic and genetic animal models have been generated for PD studies. In the neurotoxic model, chemicals such as 6-OHDA, MPTP, rotenone, paraquat, etc. are commonly used to induce PD-like symptoms in animals [9,32,42,47]. However, each of the chemical inductions has its own advantages and disadvantages, as shown in Table 1. For instance, MPTP induce PD-like pathology by targeting the cell mitochondria and thus it is a useful model to study mitochondrial dysfunction in Parkinsonism [39]. However, the MPTP method does not fully recapitulate human PD pathology in animals [86]. In the rotenone model, key features of Parkinsonism have been induced including motor deficits, catecholamine depletion, Nigral dopamine cell loss, and most importantly Lewy body formation [48,49]. As such, rotenone model is an ideal model for the study of Lewy body formation in PD pathology as compared to MPTP model.

Likewise, only certain transgenic models displayed typical PD phenotypes, for example LRRK2 and G309D-PINK1 knockout mouse model [64,78]. However, Parkin, DJ-1, and α-Synuclein knockout failed to generate the PD pathology in mouse models. Cellular-based and non-mammalian species models have been used to study the association linking genetic mutation to PD and have provided valuable insights on the molecular mechanism [59,67,70]. New animal models have also been generated to study the role of immunity in PD [85]. Unfortunately, to date, there is no perfect animal model for PD research. It is challenging to develop a model that can fully recapitulate the features of human PD, which usually take years to manifest. Despite the limitations, current animal models do provide a useful platform to selectively study the pathophysiology and the interactions of the multiple etiologic factors involved in PD.

## Figures and Tables

**Table 1 ijms-21-02464-t001:** Summary of the advantages and limitations of animal model, neurotoxic model, and genetic model in Parkinson’s Disease research.

Animal/Cell-Based Models	Advantages	Limitations	References
**Rodents**	Exhibit PD-like phenotype Established behavioral test Availability of non-motor symptoms examination Ease of genetic manipulation process	Relatively expensive Long Life Cycle	[8,9,10,11,12,67]
**Non-Human Primates (NHP)**	Close similarity in genetic and brain anatomy to human Availability of disease assessment	Laborious Very expensive Long life cycle Complicated genetic manipulation process Ethical considerations	[18,87]
***Caenorhabditis* (*C.*) *elegans***	Ease of genetic manipulation Short life cycle Low cost of maintenance Well defined neuropathology and behavior	Lack of a-Synuclein expression Difficult to target dopaminergic neuron Neuronal connectivity differs from human	[19]
**Drosophila**	Availability of transgenic model Similar dopamine synthesis pathway Exhibit PD-like phenotype	Lack of a-Synuclein homolog Limited cell death effectors	[22,88]
**Zebrafish**	Well characterized dopaminergic neuron Exhibit PD-like motor symptoms Close genetic similarity	Genetic and genomic research in progress	[23]
**iPSC model**	Ease of genetic manipulation Conduct study on patient’s cells Quick and cost effective Suitable for large scale screening	Lack of complete physiological connection that mimic brain Suitable for molecular study	[25,26,27]
**Neurotoxic Models**	**Advantages**	**Limitations**	**References**
**6-OHDA**	Able to induce massive destruction of dopaminergic neuron in SN Able to induce major behavioral deficits seen in PD	Lack of blood brain barrier penetration Acute effect Lacks LB formation	[32,33,36]
**MPTP**	Can penetrate blood brain barrier Similar topographic pattern of dopaminergic cell loss Decreased striatal dopamine levels	Lacks LB formation	[36,38,39,40]
**Paraquat**	Induce age-dependent dopaminergic neuronal loss Induce LB formation	Lack of striatal dopamine loss in some models	[42,89,90]
**Rotenone**	Behavioral impairment Dopaminergic neurodegeneration	Low reproducibility Acute toxicity	[48,91]
**Genetic Models**	**Advantages**	**Limitations**	**References**
**α-Synuclein **	∙ Useful to study α-Synuclein related degeneration To study the association between genetic and environmental factor in PD	No significant dopaminergic neuron loss Exhibit different topography pattern of cell loss	[59,60,61]
**LRRK2 **	Therapeutic target and useful for LRRK2 targeted drug test Useful for LRRK2 functional study	No significant dopaminergic neuron degeneration Lacking α-Synuclein inclusions	[63,64,65]
**Parkin **	Useful for Parkin functional study	Lack of important phenotype of PD	[67]
**DJ-1 **	Useful to combine with neurotoxin models	Lack of nigral neurodegeneration Lack of inclusion bodies	[71,72]
**PINK-1 **	Dopamine reduction and decreased locomotor activity in G309D-PINK1 mice and drosophila Useful to study the association of PINK-1/Parkin pathway in PD	Most PINK-1 models do not show reduction of dopaminergic neuron and dopamine levels.	[72,78]

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
