# Peer review of "Historical Perspective: Models of Parkinson’s Disease"

_ijms, 2020, doi:10.3390/ijms21072464_

Round 1
Reviewer 1 Report
In general, the review article makes a good impression, the work is well written, has a clear design and is well structured. However, the title of does not fully reflect its content, becouse the specific description of different animal models in the work is given only in general terms and the data shown during the administration of neurotoxins and genetic madels in different animals are not always presented in the same text volume. The review contains quit a lot of very olf references and proportion of more modern references (last 10 years) could be larger.
Author Response
Address to reviewer 1:
1. In general, the review article makes a good impression, the work is well written, has a clear design and is well structured. However, the title does not fully reflect its content, because the specific description of different animal models in the work is given only in general term
Reply: Thank you for the constructive feedback on the article title. The title of the article has changed to Historical perspective: Models of Parkinson’s disease, (Line 2) which closely reflect the contents.
2. Data shown during the administration of neurotoxins and genetic models in different animals are not always presented in the same text volume.
Reply: Additional relevant contents have added to the subcategory namely 6-OHDA and DJ-1. Following contents have added to 6-OHDA – “Numerous studies have conducted on 6-OHDA model to examine the neuroprotective function of some compounds. For instance, combined treatment of antioxidant and iron chelators have shown positive effects in neutralizing 6-OHDA neurotoxicity[34]. In addition, therapeutic effect of Tanshinone I have also been examined using the 6-OHDA model[35]. It is known that 6-OHDA model does not mimic the pathology of PD perfectly which lacks the Lewy Bodies (LB) formation[36].” (Line 153 – 157)
Following contents have added to DJ-1 – “It is important to note that DJ-1 knockout mice model is hypersensitive towards the insults of MPTP and rotenone. Locomotor of mouse model remains unaffected in the absence of DJ-1. However, apparent locomotor deficit is observed in DJ-1 knockout mice upon neurotoxin treatments as compared to WT mouse model. This indicates that DJ-1 plays a neuroprotective role in our neural system[74]. DJ-1 knockout mice could be a valuable tool to study the molecular mechanism of PD.” (Line 284-288)
3. The review contains quite a lot of very old references and proportion of more modern references (last 10 years) could be larger.
Reply: We agree with the suggestions provided. So, numerous references have been replaced with recent publication. Changes have been made to reference no.2 (Line 353-356), no.7 (Line 369-372), no.30 (Line 433-437), no.46 (Line 485-488), no.48 (Line 493-497), No.56 (Line 518-524), No.57 (Line 525-531), no.61 (Line 543-548), no.66 (Line 559-564), no.68 (Line 567-573), no.70 (Line 576-582), and no.72 (Line 586-591). We have also deleted reference no.69 (Line 574-575) because the content can be supported by reference no.68.
Reviewer 2 Report
This manuscript by Chia et al. describe variety of animal models to study Parkinson’s disease (PD) for understanding the pathogenesis and treatment development. The manuscript is a review type. Parkinson’s disease (PD) is the most common movement disorder with multifaceted features. Authors collect and provide some insight to for the potential management and treatment for PD in the animal and cell model from public databases. This retrospective article allows other scientists to quickly understand the current progress of PD and conduct in-depth research. The manuscript was carried out carefully and has completed the writing describes the techniques and findings clearly. Therefore, the manuscript is sufficient to publish in the journal.
Author Response
Address to reviewer 2:
This manuscript by Chia et.al describe variety of animal models to study Parkinson’s disease (PD) for understanding the pathogenesis and treatment development. The manuscript is a review type. Parkinson’s disease (PD) is the most common movement disorder with multifaceted features. Authors collect and provide some insight to for the potential management treatment for PD in the animal and cell model from public databases. This retrospective article allows other scientists to quickly understand the current progress of PD and conduct in-depth research. The manuscript was carried out carefully and has completed the writing describes the techniques and findings clearly. Therefore, the manuscript is sufficient to publish in the journal.
Reply: We are very thankful for your time to evaluate our review article. We appreciate the kind comments that stated in the review report.
Round 2
Reviewer 1 Report
Authors are recommended grammatical and stylistic correction of the article text in native speakers.
Author Response
Dear Editor,
We thank the reviewers for their comments and suggestions. The manuscript has been revised accordingly with Tracked Changes.
Attached is a summary of the changes.
Thanks and best regards,
Yin-Xia Chao
Reviewer suggestion: Authors are recommended grammatical and stylistic correction of the article text in native speakers.
Responses:
- Abstract (line 10-26) has been revised as “Parkinson’s disease (PD) is the most common movement disorder with motor and nonmotor signs. The current therapeutic regimen for PD is mainly symptomatic as the etio-pathophysiology has not been fully elucidated. A variety of animal models has been generated to study different aspects of the disease for understanding the pathogenesis and therapeutic development. The disease model can be generated through neurotoxin-based or genetic-based approaches in a wide range of animals such as non-human primates (NHP), rodents, zebrafish, Caenorhabditis (C.)elegans, and drosophila. Cellular-based disease model is frequently used because of the ease of manipulation and suitability for large-screen assays. In neurotoxin-induced models, chemicals such as 6-OHDA, MPTP, rotenone, and paraquat are used to recapitulate the disease. Genetic manipulation of PD-related genes, such as SNCA, LRRK2, PINK1, PRKN and DJ-1 Are used in the transgenic models. Emerging model that combines both genetic- and neurotoxin-based methods has been generated to study the role of immune system in the pathogenesis of PD. Here, we discuss the advantages and limitations of the different PD models and their utility for different research purposes.”
- Line 30-31: Deleted “changed by” and added in “named after James Parkinson who reported the clinical syndrome more than two centuries ago”.
- Line 34: Deleted “that have been described”.
- Line 38-39: Deleted “Like the aetiology, heterogenous clinical phenotype is often observed in PD patients”.
- Line 46-51: The sentence has been rephrased as “Post-mortem neuropathological confirmation is the gold standard for confirming the diagnosis. The lack of access to human brains have led scientists to develop diverse range of experimental models using animals and in vitro cultured cells that could mimic different aspects of PD.”
- Line 58-61: The sentence has been rephrased as “Rodents are commonly used as the animal model because of the ease of care in laboratory environment and availability of transgenic mouse strain as well as established protocols. Specifically, rodents are used to model PD because the nigrostriatal dopaminergic degeneration directly correlates with motor deficits observedthese animals. ”
- Line 75-93: The two paragraphs have been rephrased as “In medical research, NHP has played a critical role in providing significant insights into the mechanism of disease because NHP is closely related to human genetically and physiologically[13]. However, studies of NHP requires extensive labour and expenses as well as ethical considerations. To date, only an estimated 10% of PD research are carried out in NHP. Due to the high cost and ethical issues, NHP studies are often performed for preclinical evaluation of therapies [14]. The parkinsonian symptoms in NHP are generated through administration of neurotoxin or viral vector carrying abnormal genes. The severity of the phenotype can be measured through a Unified Parkinson’s Disease Rating Scale (UPDRS)-like measure, however this assessment on NHP has not been standardized[15].
NHP modelsexhibit symptoms like human, for instance macaques show Levodopa-induced dyskinesia that resemble human chorea and dystonia. Akinesia and axial rigidity could be assessed in NHP through jumping in the tower test and hourglass test respectively[16]. In addition, it has been suggested that macaques better replicate human sleeping pattern as compared to rodents, which makes it a superior model to study sleep or social behaviour related symptoms[17]. Moreover, neuroimaging studies have been done in NHP and that have provided valuable information when compared to patients in clinical studies [18]. Hence, NHP is a valuable animal model for PD but the study requires high resources support, expertise and is time consuming. “
- Line 99: replaced “comprises of 302 neurons” with “comprising 302 neurons”.
- Line 103: Removed “the”.
- Line 206: removed “reportedly”.
- Line 112-116: Rephrased the sentences as “Recently, Yamanaka et al made a significant breakthrough by inducing pluripotent stem cells. They generated IPSC by overexpressing four major transcription factors, OCT4, Sox2, Klf4 and C-myc[24]. iPSC-derived PD models offer an unique advantage as experiments could be performed directly on the cells isolated from patients.”
- Line 119: Replaced “is” with “are”.
- Line 121: Changed “model” into “models”.
- Line 121-123: Rephrased the sentence as “Cellular modelsare ideal for large scale drug screening that could help narrow down potential drug targets for further validation in animal models[27].”
- Line 127-134: Rephrased the sentences as “The neurotoxin-based models could be developed by introducing neurotoxins such as 6-hydroxydopamine (6-OHDA), 1-methyl-4-phenyl-1,2,3,6-tetrahydropyridine (MPTP), paraquat, and rotenone. With the addition of neurotoxins, oxidative stress is generated and this can lead to cell death in DA neuronal population. However, the disadvantage is lack the formation of Lewy bodies the main pathology hallmark of PD. Despite the limitations, these animal models have contributed significantly to discovering the disease processes and potential therapeutic targets in PD[28].”
- Line 135: Replaced “are” with “have been”.
- Line 151: Replaced “is” with “was”.
- Line 151-153: The sentence have been rephrased as “It has been widely used in PD research because of its consistent behavioral phenotype in these models and predictable degeneration in dopaminergic neurons.”
- Line 155: Replaced “rapidly oxidized and produced reactive oxygen species” with “rapidly oxidizes and produces reactive oxygen species”.
- Line 162: Added “been”; changed “model” into “models”.
- Line 163: Replaced “antioxidant” with “antioxidants”.
- Line 178: Replaced “Parkinsonism” with “PD”.
- Line 178-182: Rephrased the sentence as “To closely reproduce PD in animal models, chronic administration of the toxin needs to be carried out over weeks which will result in continuous cell death after completion of toxin administration.”
- Line 201-202: Rephrased the sentence as “For instance, experiments combining neonatal iron exposure with adult paraquat exposure hve been conducted and demonstrated age-dependent nigral dopamine cell loss[45].”
- Line 204: Replaced “compare” with “compared”.
- Line 216: Removed “there are”.
- Line 219-220: Rephrased the sentence as “This model has been used in a research studies that examined the efficacy of melatonin in neuroprotection[49].”
- Line 228: Replaced “found” with “that is present”.
- Line 229: Revised “is remain uncharacterized” into “remains uncharacterized”.
- Line 231: Removed “the”; changed “has” into “have”.
- Line 233: Changed “Lewis” into “Lewy”.
- Line 235: Removed “the group of”.
- Line 237: Replaced “Even so” with “However”.
- Line 238: Replaced “is” with “was”.
- Line 240: Replaced “is” with “was”.
- Line 241: Removed “in a nutshell”.
- Line 242: Replace “do not replicate truly the human pathology” with” did not fully replicate human pathology”.
- Line 242: Replaced “model have been generated” with “models were generated”.
- Line 246: Removed “There is an exception that”; added “However, in a model”.
- Line 247-248: Removed “has shown apparent” and added “were reported”; Replaced “is” with “was”.
- Line 253-257: Sentences have been revised to “LRRK2 mutations are associated with autosomal dominant pattern of inheritance in familial PD, with varying penetrance in different population. G2019S and R1441C/G are the two most common LRRK2 mutations[62]. ”
- Line 258: Deleted “A minor milestone has been achieved with” and added in “Interestingly”.
- Line 261: Replaced “further” with “other”.
- Line 266: Replaced “solve” with ”unravel”.
- Line 273-274: Replaced “be arising from” with “lead to”; removed “following that”.
- Line 276: Replaced “display the favourable” with “recapitulate typical”.
- Line 277-278: Rephrased the sentence as “Nonetheless, pathophysiologic studies of Parkin and its role in neuroprotection have been widely reported.”
- Line 281-282: Removed “either overexpressing a-synuclein or” and reference 69 from the sentence.
- Line 283: Replaced “it” with “Parkin itself”.
- Line 285-287: Rephrased the sentences as “DJ-1 mutations have been known to be associated with recessive forms of familial parkinsonism. Several studies suggest that DJ-1 is an antioxidant protein, which is useful in counteracting the oxidative environment of DA neurons[70]”
- Line 292: Replaced “Based on the observations, we…” with “The observations”.
- Line 296-300: Inserted paragraph “It is important to note that DJ-1 knockout mice model is hypersensitive towards the insults of MPTP and rotenone. Locomotor of mouse model remains unaffected in the absence of DJ-1. However, apparent locomotor deficit is observed in DJ-1 knockout mice upon neurotoxin treatments as compared to WT mouse model. This indicates that DJ-1 plays a neuroprotective role in our neural system[74]. DJ-1 knockout mice could be a valuable tool to study the molecular mechanism of PD.”
- Line 302-304: Rephrased the sentence as “Msutation in Pten-induced kinase 1(PINK1) are associated with recessive parkinsonism. PINK1 is a neuroprotective kinase found primarily in the mitochondria and cytosolic compartments and plays a role in neuronal differentiation[75].”
- Line 305: Deleted “Reportedly”.
- Line 306: Revised “study” to “studies”.
- Line 309: revised “show” to “showed”.
- Line 309-310: Replaced “is the main culprits of” with “was responsible for”.
- Line 311: Sentence has been revised as “Both PINK1 and parkin mutants have been found to have similar phenotypes”
- Line 314: Replaced “justify” with “support”.
- Line 315: Replaced “has rescued” with “was able to rescue”.
- Line 319: Revised the sentences as “PINK1 knockout mouse models have also been generated,”
- Line 320: Replaced “Study has shown…” with “Studies have shown…”; replaced “is ”with “was”.
- Line 323: Revised “has induced” to “can induce”.
- Line 326-327: Revised the sentence to “In vitro midbrain organoids have been generated from iPSC with different protocols[80-82].”
- Line 331: Replaced “to” with “with”.
- Line 337: Added “neurotoxic and genetic”.
- Line 337-338: Removed “which classified into the neurotoxic model and genetic model”.
- Line 343: Replaced “wholly replicate the” with “fully recapitulate”.
- Line 345 & 346: Revised “Lewis” to “Lewy”.
- Line 348: Replaced “favourable” with “typical”; revised “phenotype” as “phenotypes”.
- Line 350: revised “model” to “models” and deleted “In recent advances”.
- Line 352: Replaced “of” with “linking”; deleted “pathology which” and added “and have”.
- Line 353-361: Rephrased the sentences as “New animal models have also been generated to study the role of immunity in PD[86]. Unfortunately, to date, there is no perfect animal model for PD research. It is challenging to develop a model that can fully recapitulate the features of human PD, which usally take years to manifest. Despite the limitations, current animal models do provide a useful platform to selectively study the pathophysiology and the interactions of the multiple etiologic factors involved in PD.”
- Line 369-371: revised the Acknowledgements part as “The authors like to thank National Medical Research Council for their support (EK-T for StaR and PD LCG 002, SPARKII, YX-C for Transition award).”